# Morphofunctional Investigation in a Transgenic Mouse Model of Alzheimer’s Disease: Non-Reactive Astrocytes Are Involved in Aβ Load and Reactive Astrocytes in Plaque Build-Up

**DOI:** 10.3390/cells12182258

**Published:** 2023-09-12

**Authors:** Daniele Lana, Jacopo Junio Valerio Branca, Giovanni Delfino, Maria Grazia Giovannini, Fiorella Casamenti, Pamela Nardiello, Monica Bucciantini, Massimo Stefani, Petr Zach, Sandra Zecchi-Orlandini, Daniele Nosi

**Affiliations:** 1Department of Health Sciences, University of Florence, 50134 Florence, Italy; daniele.lana@unifi.it (D.L.); mariagrazia.giovannini@unifi.it (M.G.G.); 2Department of Experimental and Clinical Medicine, University of Florence, 50134 Florence, Italy; jacopojuniovalerio.branca@unifi.it (J.J.V.B.); sandra.zecchi@unifi.it (S.Z.-O.); 3Department of Biology, University of Florence, 50121 Florence, Italy; giovanni.delfino@unifi.it; 4Department of Neuroscience, Psychology, Drug Research and Child Health, University of Florence, 50134 Florence, Italy; fiorella.casamenti@unifi.it; 5General Laboratory, Careggi University Hospital, 50134 Florence, Italy; pamela.nardiello@unifi.it; 6Department of Experimental and Clinical Biomedical Sciences, University of Florence, 50134 Florence, Italy; monica.bucciantini@unifi.it (M.B.); massimo.stefani@unifi.it (M.S.); 7Department of Anatomy, Third Faculty of Medicine, Charles University, 100 00 Prague, Czech Republic; petr.zach@lf3.cuni.cz; 8DMSC Imaging Platform, 50134 Florence, Italy

**Keywords:** neuroinflammation, neurodegeneration, hippocampus, glial cells, cell–cell interactions, clasmatodendrosis, amyloid plaques, Aβ-aggregates, transgenic mouse, confocal microscopy

## Abstract

The term neuroinflammation defines the reactions of astrocytes and microglia to alterations in homeostasis in the diseased central nervous system (CNS), the exacerbation of which contributes to the neurodegenerative effects of Alzheimer’s disease (AD). Local environmental conditions, such as the presence of proinflammatory molecules, mechanical properties of the extracellular matrix (ECM), and local cell–cell interactions, are determinants of glial cell phenotypes. In AD, the load of the cytotoxic/proinflammatory amyloid β (Aβ) peptide is a microenvironmental component increasingly growing in the CNS, imposing time-evolving challenges on resident cells. This study aimed to investigate the temporal and spatial variations of the effects produced by this process on astrocytes and microglia, either directly or by interfering in their interactions. *Ex vivo* confocal analyses of hippocampal sections from the mouse model TgCRND8 at different ages have shown that overproduction of Aβ peptide induced early and time-persistent disassembly of functional astroglial syncytium and promoted a senile phenotype of reactive microglia, hindering Aβ clearance. In the late stages of the disease, these patterns were altered in the presence of Aβ-plaques, surrounded by typically reactive astrocytes and microglia. Morphofunctional characterization of peri-plaque gliosis revealed a direct contribution of astrocytes in plaque buildup that might result in shielding Aβ-peptide cytotoxicity and, as a side effect, in exacerbating neuroinflammation.

## 1. Introduction

The term “neuroinflammation” was originally coined to describe the accumulation of leukocytes in the degenerating white matter and blood vessels in multiple sclerosis and was later extended to the abnormal crowding of reactive microglia around amyloid plaques in Alzheimer’s disease (AD) [1]. Microglia, while also involved in nerve tissue maintenance, represent the immunocompetent cell line of the central nervous system. In AD, the overproduction and fibrillar aggregation of denatured Aβ-peptides and the build-up of amyloid plaques induce the release of pro-inflammatory cytokines by neurons [2,3], thus promoting the immune response of microglia [4,5]. The concept of neuroinflammation has been then applied to numerous neurodegenerative pathologies, such as Parkinson’s disease (PD), acute CNS injuries, and some psychiatric disorders, and included further cell types, such as astrocytes. In AD, astrocytes are involved in Aβ-peptide clearance [6,7] and modulate microglial phagocytosis [4,8,9,10,11]. Although the primary concept of neuroinflammation refers to a neuroprotective response of non-neuronal cells reacting to alterations in central nervous system (CNS) homeostasis, its chronicity may also involve neurodegenerative aspects. In AD, this has been correlated to dysfunctional alterations in the phenotype of reactive microglia, resulting in increased production of amyloid fibrils [12,13,14] and phagocytosis of healthy neurons [15,16,17]. In this view, two opposite functional classifications of reactive microglia have been introduced: the M1-neurodegenerative/proinflammatory phenotype and the M2-neuroprotective/anti-inflammatory phenotype [18]. However, transcriptome studies in different models of neuroinflammation indicate the simultaneous expressions of both neurodegenerative and neuroprotective factors by microglia [19]. Therefore, M1 and M2 should be proposed, more properly, as conjectural opposites in a wide range of reactivity states. Morphological traits of microglia dysfunction during neuroinflammation consist of loss of cytoplasmic processes and elongation of the cell body, resulting in amoeboid morphology [20]. In the normal CNS, microglial cytoplasmic processes are numerous, thin, and highly branched: they extend from the cell body, acting as sensors in the surrounding microenvironment [4]. In neuroinflammation, the branching of microglial cytoplasmic processes is oriented towards pro-inflammatory molecules, following chemotactic and mechanotactic signals, thus playing a key role in phagocytosis [4,10].

It is commonly accepted that reactive microglia induce reactivity of astrocytes [21], characterized by hypertrophic morphology and increased expression of the astrocyte marker glial fibrillary acidic protein (GFAP) [22] in astrocyte cytoplasmic projections (APJs). Reactive astrocytes may, in turn, undergo neurodegenerative phenotype changes in AD chronic neuroinflammation [23,24,25], likely induced by pro-inflammatory microglia [24]. However, these reciprocal induction exchanges do not clarify whether microglia, although being the first glial cell line to “react” to Aβ-peptide, are also the first ones to “sense” it. Actually, this hypothesis appears questionable when considering some morphofunctional traits of protoplasmic astrocytes in gray matter. Indeed, APJs are numerous and highly ramified, forming a dense, spongious meshwork throughout the gray matter. APJs express high levels of connexins, among which connexin 43 (Cx43) is the predominant one, enabling them to establish reciprocal gap junctions. By means of such connections, astrocytes behave as a functional syncytium [26] and play structural and maintenance roles in the CNS [12]. It is worth noting that the astrocyte endfeet of glia limitans wrap around blood vessels and need Cx43 to facilitate the entry of plasma ultrafiltrate into the neuropil. Indeed, astrocytes may also be involved in plasma ultrafiltrate diffusion throughout the parenchyma from arteries to veins, thus establishing a glymphatic circulation within the tissue [27]. In addition, astrocytes directly interact with neurons and microglia through Cx43-mediated gap junctions [28] and β1 integrin-mediated cell–cell contacts [10], respectively.

These considerations suggest that contacts of APJs with Aβ-peptides are early events in the pathogenesis of AD, thus raising the question of whether Aβ may affect the activities of astrocytes prior to the onset of their reactivity. Reasonably, such a hypothesis could also affect the microglial response due to the constitutive, mutual interactions between these two cell types [12]. In our previous studies on animal models of CNS aging, it was demonstrated that Aβ-peptide deposition induces fragmentation of APJs (clasmatodendrosis) and a consequent decrease in APJ-mediated clearance activity [29]. In the same models, clasmatodendrosis underlay the disruption of the astrocyte meshwork, impairing the branching of microglia and, in turn, their clearance activity [10]. Actually, clasmatodendrosis involves a decrease in astrocyte expression of GFAP and can be classified as a passive, non-reactive change due to altered environmental conditions [30] rather than a reactive response of astrocytes.

Considering that the sub-chronic inflammation frequently occurring in the aged CNS represents a prodrome of AD [13], the first aim of this study was to verify the presence and role of clasmatodendrosis in a transgenic mice model of Aβ-deposition (TgCRND8). Indeed, a working hypothesis of this study is that astrocytes engage in early interactions with Aβ peptides. The consequent induction of clasmatodendrosis could, therefore, affect astrocytes, leading to increased Aβ-loading in this AD model as well. Likewise, microglial clearance may be impaired by reduced contact interactions between APJs and microglial branches. High-resolution confocal analyses were performed in mice at three stages of the disease, namely: 2 months old mice (2m-Tg-m), characterized by mild cognitive impairment in the absence of amyloid plaques; 6 and 12 months-old mice (6m- and 12m-Tg-m), representing the intermediate and advanced stages of the disease, respectively.

Although the association of amyloid plaques with distinctive peri-plaque gliosis is known, the role played by APJs involved in this reactive response has not been adequately investigated. The debate focuses on whether such an association exerts neuroprotective or neurodegenerative effects [31]. On the one hand, peri-plaque gliosis may play a barrier-like role similar to perivascular gliosis. Moreover, astrocytes express Aβ degrading enzymes [32,33] and can clear small deposits of Aβ fibrils [29,34], although it is unclear whether these enzymes are released in peri-plaque gliosis. On the other hand, it has been suggested that astrocytes express Aβ-peptide and contribute significantly to Aβ deposition [35]. This neurodegenerative vs. neuroprotective paradigm also involves the growth of amyloid plaques. In the APP/PS1 AD mice model, a positive correlation was found between gliosis and Aβ-plaque growth by in vivo multiphoton microscopy [36], whereas this correlation was negative in *ex vivo* investigations after attenuation of astrocyte reactivity [37]. Finally, besides M1 and M2, a definite transcriptomic signature (A1) has been suggested to characterize neurotoxic astrocytes in AD [23]. Supporting this hypothesis, saturated lipids contained in apolipoproteins E (APOE) and J (APOJ) were identified as probable astrocyte-derived neurotoxic factors [38]. Moreover, the involvement of APOE4, an isoform of the APOE family, in Tau-mediated neurodegeneration [39] and constitutive expression of the peptide Aβ [40] has been recently suggested. However, the persistent homeostatic instability and intensification of astrocyte activity may, *per se*, result in the impairment of relevant functions, such as glutamate uptake [41,42], K+ buffering [43], and GABA modulation [44]. In addition, the energy supply to neurons becomes inadequate [45], entailing the production of harmful fatty acid metabolites and reactive oxygen species [46] as a further maladaptive side effect.

Given this fragmentary information, a further aim of this study was to broaden the current knowledge on the morphological and functional features of astrocytes involved in peri-plaque gliosis in order to obtain new insights on their role in amyloid plaque build-up.

## 2. Materials and Methods

### 2.1. TgCRND8 Mice

The TgCRND8 mice (Tg-m) express two mutated human amyloid precursor protein (APP) genes implicated in AD (Swedish, KM670/672NL and Indiana, V717F) under the regulation by the Syrian hamster prion promoter gene [47]. The mice were maintained on a hybrid (C57)/(C57/CH3) background by crossing transgenic heterozygous Tg-m males with wild-type (WT) female mice.

The two mutations involve both the β and γ secretase APP cleavage sites [47,48]. Aβ induced cognitive impairment in this model occurs before 3 months of age and the formation of amyloid plaques [49]. The transgenic mice were generated and supplied by Dr. P. St George Hyslop (Center for Research in Neurodegenerative Diseases, Toronto, ON, Canada), and the colony was bred in the inter-departmental animal house facility (Ce.S.A.L., Centro Stabulazione Animali da Laboratorio) of the University of Florence. All experiments on animals were performed according to the Italian Law on Animal Welfare (DL 26/2014) upon prior approval by the Animal Care and Use Committee of the University of Florence and the Italian Ministry of Health. All efforts were made to minimize animal suffering and to use the minimum number of animals needed to obtain statistically reliable data. Three groups of transgenic mice, equally divided for sex, were used: 2 months old (2m-Tg-m, *n* = 6), 6 months old (6m-Tg-m, *n* = 6), and 12 months old (12m-Tg-m, *n* = 6). Age-matched WT mice (WT-m), equally divided for sex, of 2, 6, and 12 months of age (*n* = 6 each) were used as controls. Since no significant differences were ever observed in any of the parameters investigated, the data from the three groups of WT-m were averaged.

At the appropriate ages (2, 6, or 12 months), mice were euthanized by deep anesthesia (Zoletil, 80 mg/kg i.p.) and transcardiac perfusion with 200 mL of ice-cold 4% paraformaldehyde in phosphate-buffered saline (PBS), at pH 7.4. After overnight post-fixation and cryoprotection (18% sucrose/PBS), 40 μm-thick whole brain coronal sections containing the dorsal hippocampus (coordinates −1.6 to −2.0 mm from bregma) were collected from both left and right hemispheres with a cryostat, and stored at −20 °C in antifreeze solution. In these sections, the *stratum radiatum* and the *stratum pyramidale* of the hippocampal CA1 region were examined to assess APJs integrity and clearance activity, respectively.

### 2.2. Immunofluorescence Staining

Immunohistochemistry was performed on free-floating sections [50]. The list of antibodies and protocols used for the different immunostainings are reported in Table 1. All the washings were performed thrice for 5 min.

#### 2.2.1. Iba1 + GFAP + Aβ Triple Labeling Immunostaining

Day 1. Free-floating sections (40 μm thick) were placed in 24 well plates, rinsed in PBS-Triton X100 (TX), and blocked for 60 min with blocking buffer (BB, 10% normal goat serum in PBS-TX and 0.05% NaN). Sections were then incubated overnight at 4 °C under slight agitation in a BB solution of two combined primary antibodies: a rabbit anti-Iba1 antibody + a mouse anti-β amyloid antibody.

Day 2. After washing, the sections were incubated for 2 h at room temperature in the dark with AlexaFluor 555 donkey anti-mouse IgG (1:400) secondary antibody diluted in BB and then for 2 h at room temperature in the dark with AlexaFluor 555 donkey anti-mouse IgG (1:400, Code #A31570, Thermo Fisher Scientific, Milan, Italy) + AlexaFluor 635 goat anti-rabbit IgG (1:400, Code #A31577, Thermo Fisher Scientific). After washing, the astrocytes were immunostained using a mouse anti-GFAP antibody conjugated with the fluorochrome AlexaFluor 488 (1:500, Code #MAB3402X, Millipore, Billerica, MA, USA).

#### 2.2.2. Iba1 + CD68 Double Labeling Immunostaining

Combined primary and secondary polyclonal antibodies were employed to optimize staining efficiency in machine learning and tracing analyses. Mixed amyloid/lipofuscin autofluorescence is also shown. The procedure was the same as above, performed with specific antibodies:

Day 1: The primary antibodies were a rabbit anti-Iba1 antibody + a mouse anti-CD68 antibody.

Day 2: The secondary antibodies were: in the first incubation AlexaFluor 488 donkey anti-rabbit (1:400, Code #A21206, Thermo Fisher Scientific), in the second incubation AlexaFluor 488 donkey anti-rabbit (1:400) + with AlexaFluor 555 donkey anti-mouse IgG (1:400).

#### 2.2.3. GFAP Single Labeling Immunostaining

Combined primary and secondary polyclonal antibodies were employed to optimize staining efficiency in machine learning and tracing analyses.

The procedure was the same as above, performed with specific antibodies:

Day 1: The primary antibody was a rabbit anti-GFAP antibody.

Day 2: The secondary antibody was a polyclonal AlexaFluor 488 donkey anti-rabbit (1:400, Code #A21206, Thermo Fisher Scientific).

#### 2.2.4. Cx43/MAP2 + GFAP Double Labeling Immunostaining

The procedure was the same as above, performed with specific antibodies:

Day 1: The primary antibody was a rabbit anti-Cx43 antibody or a rabbit anti-MAP2 antibody.

Day 2: The first incubation with secondary antibody AlexaFluor 635 goat anti-rabbit (1:400), second incubation with anti-GFAP antibody conjugated with the fluorochrome AlexaFluor 488, dilution (1:500).

After washing, the sections treated with the three different immunostainings were mounted onto gelatin-coated slides using an anti-fading Vectashield mounting medium supplemented with DAPI for nuclear counterstaining (Vector Laboratories, DBA ITALIA S.R.L., Milan, Italy).

### 2.3. Confocal Microscopy

Samples were analyzed by using either a Leica TCS-SP5 or a Leica Stellaris 5 confocal microscope equipped with a Plan Apo 63X Oil Immersion objective (NA = 1.40). A series of confocal stacks (pixel size 150 nm, Z-steps 200 nm) were acquired through the depth of sections. Either previously described methods of spectral unmixing [29] or Tau Sense separation were used to isolate autofluorescence from immunofluorescence. Mixed autofluorescence of misfolded peptides, including the Aβ aggregates and lipofuscins (referred to as denatured peptides in figures), was also obtained by spectral unmixing techniques as previously reported [29] to detect amyloid plaques and deposits. Different color combinations were employed case-by-case to optimize the visualization of the diverse markers and cell components in the figures. The size of the volumes of interest (VOIs) in 3D analyses was determined case-by-case to optimize data reliability on the different structures/processes.

#### 2.3.1. Marker Expression Analyses

Analyses of fluorescence from the different immunostainings were performed in the CA1 region of 6 coronal sections of the hippocampus per mouse. Five optical volumes (246 µm × 246 µm × 30 µm) were randomly acquired in each section. Quantitative analyses of the expression of the different markers were performed by using Fiji software (version 2.9.0/1.53t) [51]. Mean intensity, as well as integrated density of the various immunolabeled markers, were quantitatively analyzed depending on the type of information needed: (i) The mean intensity of fluorescence provides information on the intracellular concentration of the fluorochrome; (ii) the integrated density corresponds to the total amount of the analyzed marker.

#### 2.3.2. Tracing Analysis of Astrocyte Processes

Analyses were performed on immunostained samples to reveal only GFAP, thus maximizing the spectral window for the acquisition of the emitted fluorescence. Morphometric measurements of APJs were performed by using the SNT (Simple Neurite Tracer) plugin in Fiji [52] on single astrocytes. APJs of at least 20 astrocytes residing in the *stratum radiatum* per animal group were traced and reconstructed. APJs were classified from the first to the fifth order on the basis of their sprout and thickness: first-order APJs were the thickest and sprouted from astrocyte soma, fifth order was the thinnest sprouting from the fourth order APJs. The total length of all APJs (µm) and their number were measured in each cell. Sholl’s analysis was also performed on representative cells to evaluate the extension and complexity of APJ arborization in each animal group [53]. Briefly, this analysis virtually creates shells (circles in 2D images, spheres in 3D stacks) of increasing diameter around the center of a cell and counts how many times the arbor of APJs intersects each shell (see Appendix A and Appendix A). GFAP+ cytoskeleton within cell bodies was omitted from measurements.

#### 2.3.3. Machine Learning Analyses

All machine learning analyses were performed using the AIVIA software (https://www.aivia-software.com; version 12.1). This software exploits the random tree algorithm to recognize (classify) distinct structures in 2D images and 3D volumetric scans by integrating various parameters of the sample structures indicated by the operator during the teaching process: signal intensity, shape, curvatures of the edges of the structures, and gradient of signal intensity from their edge to the outside. Each classified structure is then mapped into a “probability map” that graphically reports its percentage probability to occupy a given area in 2D images or volume in 3D stacks. Integrated analysis tools then allow the classified structures to be segmented to assess their signal intensity (mean intensity, integrated density, standard deviation, max, min, etc.) and morphometric parameters (number, volume, area, elongation, etc.).

##### Autofluorescent Deposits in the CA1 *Stratum Pyramidale*

Single immunostainings were performed to reveal GFAP. At least 10 VOIs (92 µm × 92 µm × 26 µm) were analyzed in at least 3 sections from each animal group. To allow discrimination of signals emitted by autofluorescent deposits from the overlapping emission of Alexa488 on the immunostained GFAP, three different channels were acquired in each 3D confocal scan (see Appendix A and Appendix A). In the first of two sequential scans, the light from two spectral windows was simultaneously collected using the 458 nm laser line: the first one was set in the 495–525 nm spectral range (495–525_l458_); the second one in the 560–590 nm range (560–590_l458_). These two channels contained both immunofluorescence and autofluorescence with a different intensity ratio; after applying an appropriate color scale to each channel, they were merged to discriminate the autofluorescent deposits. A third channel was sequentially scanned in the 560–590 nm spectral window using the 488 laser line (560–590_l488_) to minimize autofluorescence and optimize Alexa488 emission. The 495–525_l458_ and 560–590_l458_ confocal scans were employed in machine MLA to obtain a new channel with only autofluorescent deposits. This new channel was used together with channels 495–525_l458_ and 560–590_l488_ to identify contacts between autofluorescent deposits and APJs. Finally, autofluorescent deposits associated with APJs were discriminated using the channels related to deposits and contacts generated by MLA and the 560–590_l488_ channel. The total number of autofluorescent deposits, their association rate with APJs (%), and the number of non-associated deposits were measured in each VOI.

##### Cx43 Clusters in CA1 *Stratum Radiatum* (Glia Limitans, Perivascular and Peri-Plaque Gliosis)

MLA was also employed in order to perform identification and morphometric analyses of Cx43 clusters associated with APJs in normal blood vessels as well as in perivascular and peri-plaque gliosis. Clusters of Cx43 were identified in volumes of interest (VOIs, 50 µm × 50 µm × 11 µm). Volume measurements were performed by the AIVIA tool 3D object analysis, and a volume threshold of 1.5 µm was chosen to select clusters in the astrocyte endfeet.

##### Dendrite Fragmentation in CA1 *Stratum Radiatum* (Peri-Plaque Gliosis)

The identification and isolation of MAP2+ dendrite fragments was performed by MLA. At least 10 VOIs (30 µm × 30 µm × 20 µm) were analyzed in at least 3 sections from each animal group. The 3D volume renderings of confocal stacks were obtained with the Horos software (https://horosproject.org; version 4.0.0) or with AIVIA software. The integrated density of GFAP+ fragments was measured.

### 2.4. Statistical Analysis

Statistical analyses were performed by using the Jamovi open-source software (The Jamovi project (2021). Jamovi (Version 1.6), retrieved from https://www.jamovi.org). One-way ANOVA analysis of variance was followed by Tukey *post hoc* tests. All values are reported as means ± SE. The parameter “*n*” reported in all statistical analyses indicates the number of animals analyzed in each animal group.

## 3. Results

### 3.1. Analysis of Astrocyte (GFAP) and Microglia (Iba1) Morphology and Localization, and of GFAP and Aβ-Peptide Expression

To frame Aβ-peptide modifications on astrocytes in a neuroinflammatory context, in situ expression of GFAP, a marker of APJs, and Iba1, a microglial marker involved in branching, migration, and phagocytosis, were analyzed by confocal microscopy in the CA1 hippocampus of 2m- 6m- and 12m-Tg-m. In fact, recent evidence from our research group indicates that the APJ meshwork provides a scaffold for cell–cell contacts with microglia, promoting their branching and migration in neuroinflammation [10]. In addition, a close correlation was found in aged rats between disassembly of the APJ scaffold due to clasmatodendrosis and inhibition of microglia branching. Aβ-peptide deposition was also revealed by immunostaining. Figure 1 shows representative Z-projections of the acquired optical volumes and the related quantitative evaluations. The mean Intensity of GFAP and integrated density of GFAP and Aβ-peptide immunofluorescence, depicting the density and expression of the labeled molecule, respectively, are reported. Of note, levels of integrated immunofluorescence density of Aβ-peptide in the *stratum pyramidale* were already significantly higher in 2m-Tg-m than in WT-m (Figure 1E). Concomitantly, GFAP immunofluorescence levels in 2m-Tg-m were lower than in WT-m (Figure 1F,G), suggesting that Aβ overproduction may induce a non-reactive modification in astrocytes. Accordingly, the APJs in transgenic mice appeared shorter (Figure 1B) and less branched than in WT-m (Figure 1A), apparently providing less support for their contacts with microglial branches (Figure 1 inner insets in A and B). In this context, microglia showed elongated bodies and lower branching in comparison with Wt-m.

In the *stratum pyramidale* of the 6m- and 12m-Tg-m, a trend of increased expression of Aβ-peptides was detected (Figure 1E). In these mice, Aβ-immunofluorescence was also concentrated in large amyloid plaques mainly located in the *stratum radiatum* (Figure 1C panel c3, 1D panel d2) and *stratum oriens*. Samples of 6m-Tg-m showed early APJ hypertrophy in reactive astrocytes contacting and encircling the small amyloid plaques (less than 20 µm in size; Figure 1C panel c2 IN). Moreover, many hypertrophic astrocytes surrounded large amyloid plaques of both 6m- and 12m-Tg-m (Figure 1C panel c3 IN, 1D panel d2 IN) and were in contact with numerous microglia, possibly involved in the phagocytosis of Aβ peptides (Figure 1C inset 1, 1D inset 2). Hypertrophic astrocytes showed significantly elevated levels of GFAP expression in these regions (Figure 1F), confirming their reactive state, while microglia showed reduced branching compared with WT-m, but their soma appeared less elongated than in 2m-Tg-m. Away from amyloid plaques, astrocytes of 6m-Tg-m showed shortened APJs and low levels of GFAP expression (*stratum radiatum*, Figure 1C panel c1 OUT, 1F and 1G). In agreement with previous findings [54], short APJs occurring in equivalent areas of 12m-Tg-m appeared thicker than in 2m-Tg-m and 6m-Tg-m and, therefore, the expression of GFAP in these mice was not significantly different from WT-m (*stratum radiatum*, Figure 1D panel d1 OUT, 1F, 1G). Usually, the microglia accompanying these non-reactive astrocytes in the 6m- and 12m-Tg-m showed amoeboid morphology, confirming a correlation between astrocytic meshwork integrity and microglial branching (Figure 1D panel d1 OUT, 1C panel c1 OUT). As a hallmark of 12m-Tg-m, hypertrophic APJs were detectable away from the plaques, forming intermingled tubular nets similar to the glial scars associated with damaged vessels (Figure 1D panel d1 OUT).

Overall, the above data suggest that Aβ-deposition induces early non-reactive modifications in astrocytes. In intermediate and advanced stages of the disease, Aβ-plaques may raise major environmental challenges for astrocytes, which are closely correlated with their phenotypic shift to the hypertrophic morphology. However, the concomitance of astrocytic meshwork disassembly and amoeboid microglial morphology in non-plaque regions appears to be a common, early, and persistent feature of the disease in this animal model.

### 3.2. Analysis of Microglial Iba1 and CD68 Expression and In Situ Localization

A previous study from our research group showed that in aged rat models, microglia located in regions of astrocytic meshwork disassembly exhibit impaired migration along with activated phagocytosis [10]. To characterize the in situ activation of both microglial phagocytosis and migration processes in this AD model, the coexpression of two microglia markers was assessed in the CA1 hippocampus of 2m-, 6m- and 12m-Tg-m by double immunostaining of Iba1 and CD68, which selectively marks the presence of phagocytic vesicles. Mixed amyloid/lipofuscin autofluorescence was also obtained by spectral unmixing. Representative Z-projections for qualitative evaluation of microglial cell morphology and cellular localization of the two fluorescent signals are shown in Figure 2, and quantitative analyses of their integrated density and mean intensity are shown in Figure 3.

At first, a comparison between WT-m and 2m-Tg-m was performed in the *stratum radiatum* to assess early modifications in the expression of Iba1 and CD68 induced by overproduction of APP. In WT-m, microglia showed numerous Iba1^+^ branches, radially emanating from the cell bodies (Figure 2A), possibly involved in intense interactions with surrounding neurons, astrocytes, and the extracellular matrix [10,12,55]. Faint CD68 immunofluorescence was detected in branches of microglia (Figure 2A, inner insets in 1 and 2), fitting their constitutive involvement in phagocytic turnover processes. In 2m-Tg-m, the development of AD-like disease was evidenced by small autofluorescent amyloid/lipofuscin deposits in the *stratum pyramidale* (Figure 2B). Prominent morphological features of microglia depicted by Iba1 staining consisted of retracted branches and elongated cell bodies (*stratum radiatum,*
Figure 2B, insets 3 and 4). Unexpectedly, the intracellular concentration of Iba1, estimated by the mean intensity of the immunofluorescent staining, did not show any significant difference between WT-m and 2m-Tg-m (Figure 3A). Moreover, evaluation of Iba1 expression by the integrated density of immunofluorescence provided significantly lower values in the latter animal group (Figure 3B), consistent with lesser branching. These data suggested that microglia in the 2m-Tg-m hippocampus were characterized by low efficiency in branch-mediated mechanisms of orientation towards proinflammatory/cytotoxic molecules. On the other hand, a slight, not significant, increase in mean intensity and a significant increase in integrated density of the phagocytosis marker CD68 were found in the 2m-Tg-m (Figure 3C,D). Indeed, high-density spots of CD68 were observed on both cell processes and bodies of these cells (Figure 2B, inner insets in 3 and 4), indicating local activation of the molecular pool for phagocytosis.

In the CA1 hippocampus of 6m-Tg-m, large deposits of autofluorescent material (*stratum radiatum*, Figure 2C panel c1 IN), corresponding to amyloid plaques (see Figure 1), were found. The crowded microglial cells around the plaques showed numerous short and nearly unbranched Iba1^+^ processes, with high levels of CD68 immunofluorescence at their ends, and were often engaged in multiple soma-to-soma adhesions. (Figure 2C panel c1 IN, inset 5). In these cells, the mean intensity of Iba1 immunofluorescence was higher than in the WT-m and 2m-Tg-m, whereas integrated density was higher than in 2m-Tg-m and lower than in the WT-m (Figure 3A,B). In parallel, both the mean intensity and integrated density of CD68 immunofluorescence in 6m-Tg-m were higher than in WT-m and 2m-Tg-m (Figure 3C,D). By contrast, in the areas of the *stratum radiatum* away from amyloid plaques, microglia appeared different while showing remarkable similarities to those of 2m-Tg-m in terms of elongated cell body and poor branching (Figure 2C panel c2 OUT, inset 6). Within such areas of 6m-Tg-m, the mean intensity of Iba1 immunofluorescence was not different from that of WT-m and 2m-Tg-m, whereas its integrated density was lower (Figure 3A,B). On the other hand, both the mean intensity and integrated density of CD68 immunofluorescence were higher than in WT-m and 2m-Tg-m (Figure 3C,D). By comparing the two different hippocampal areas of these mice, it was found that microglia surrounding amyloid plaques showed higher levels of both Iba1 and CD68 markers than far away (Figure 3).

Large autofluorescent deposits were numerous in 12m-Tg-m (*stratum radiatum*, Figure 2D) and surrounded by small-sized microglia with short processes (Figure 2D panel d1 IN), expressing high levels of both Iba1 and CD68 (Figure 2D and Figure 3A–D). In the same areas, patterns of soma-to-soma adhesion were also found among microglial cells, characterized by high concentrations of both Iba1 and CD68 at sites of closer intercellular contacts (Figure 2D insets 7 and 7a), indicating active phagocytosis. Far away from amyloid plaques, microglia in 12m-Tg-m showed higher values of both mean intensity and integrated density of Iba1 fluorescence in comparison with similar areas of 6m-Tg-m (Figure 3A,B). However, Iba1 integrated density in these microglia was still significantly lower than in WT-m (Figure 3B). Accordingly, microglia with intermediate levels of branching (Figure 2D panel d2 OUT) were observed along with numerous unbranched microglia (Figure 2D panel d2 OUT, insets 8 and 9).

The above results suggest that the amyloid plaques in this mouse model identify specific environmental microdomains in which the high expression levels of Iba1 and CD68 of microglia, both in the intermediate and late stages of the disease, correspond to a typical reactive state. However, at all stages of the disease, microglia located away from the plaque microdomains, i.e., in regions of astrocyte network disassembly, show a distinctly different reactive, senile-like phenotype, consistent with altered branching and hence migration processes.

### 3.3. Morphological and Morphometric Analyses of GFAP+ APJs in the Stratum Radiatum and MLA Assessment of Their Interactions with Lipofuscin/Aβ–Deposits in the Stratum Pyramidale

To quantitatively assess the previously observed disassembly of the astrocytic meshwork, morphological and morphometric analyses of GFAP+ APJs were performed in non-plaque microdomains of the *stratum radiatum* at different stages of the disease. A qualitative comparison of APJ morphology in plaque and non-plaque microdomains was also performed. In addition, our previous studies have shown that increased Aβ-peptide loading in the CA1 hippocampus of aged rats results in clasmatodendrosis, widespread disassembly of the astrocytic meshwork, and impaired clearance activity of astrocytes in the *stratum pyramidale* [29]. Therefore, MLA analysis of the rate of association between APJs and autofluorescent deposits, expressing the efficiency of astrocyte clearance activity [29], was performed in the *stratum pyramidale* of Tg-m. Single labeling immunostaining was performed to optimize GFAP immunofluorescence intensity and detection in tracing and learning machine analyses. Mixed autofluorescence from lipofuscin/Aβ-peptide deposits is also shown. Representative z-projections for qualitative and morphological assessment of astrocyte morphology in the *stratum radiatum* and *stratum oriens* are shown in Figure 4, while morphometric and Sholl’s analyses of APJs in the *stratum radiatum,* as well as MLA of APJ interactions with autofluorescent deposits in the *stratum pyramidale,* are shown in Figure 5.

In WT-m, APJs of the first, second, and third branching orders were evenly distributed in the *stratum radiatum* of CA1 (Figure 4A) and were characterized by regular course and gradual proximal-distal thinning (Figure 4A and inset 1). Fourth and fifth-order APJs were detected at higher magnification, deeply infiltrating the gray matter (Figure 4A inset 2). In 2m-Tg-m, the APJ meshwork exhibited clear alterations: first-to-third order processes were frequently irregularly shaped due to spiral twisting and sharp disruptions (*stratum radiatum*, Figure 4B). Discrete fragments of APJs were also evident, indicating the occurrence of clasmatodendrosis (Figure 4B insets 3 and 4). Moreover, fourth and fifth-order APJs were highly reduced in number (Figure 4B inset 4).

Autofluorescent amyloid plaques were detected in 6m-Tg-m (*stratum oriens*, Figure 4C panel c1 IN) and, in agreement with known data, these plaques were surrounded by numerous, long and thick APJs showing the hypertrophic morphology typical of the reactive status (Figure 4C panel c1 IN and inset 5). Of note, APJs close to the amyloid plaques formed intricate and thick nets that hindered recognition of their actual shape and direction, likely due to the presence of numerous GFAP+ fragments (Figure 4C inset 6). On the other hand, astrocytes in non-plaque microdomains showed scanty APJs (*stratum radiatum*, Figure 4C panel c2 OUT) and evident abnormalities, including fragmentation of first (Figure 4C inset 7) and second-order APJs (Figure 4C inset 8).

In 12m-Tg-m, a high expression of GFAP in APJs was a predominant trait of astrocytes (Figure 4D panels d1 IN and d2 OUT); in particular, hypertrophic astrocytes located in plaque-microdomains adhered to the plaques by their APJs (*stratum radiatum*, Figure 4D panel d1 IN). The presence of stubby primary APJs surrounded by numerous GFAP+ fragments, typical of clasmatodendrosis (Figure 4D insets 9 and 10), was suggestive of the noxious effects of these cell-plaque contacts. Consistent with previous data [56], the thick APJs of dystrophic astrocytes in non-plaque microdomains showed a low degree of branching (*stratum radiatum*, Figure 4D panel d2 OUT and inset 11). In these cells, the primary APJs often appeared spiral-shaped (Figure 4D inset 11), while the third- and fourth-order APJs formed a loose meshwork (Figure 4D insets 11 and 12), indicating local astrocyte damage.

Then, quantitative three-dimensional evaluations of APJ branching were performed to assess in terms of the number and total length of APJs the extent of the above-described morphological changes in the Tg-m (Figure 5). These analyses were limited to non-plaque microdomains because of the reported difficulty in discriminating APJs in the meshwork around plaques. It was found that both the total length and the number of APJs were lower at any stage of disease in Tg-m than in WT-m (Figure 5A,B). By Sholl’s analysis, the arborization of APJs in 2m-, 6m- and 12m-Tg-m astrocytes showed both a reduced number of intersections with Sholl’s spheres and a lesser extent, as compared with WT-m astrocytes (*stratum radiatum*, Figure 5C–F, Appendix A, and Appendix A). These data indicated a reduction in both density and volume of infiltration of APJs in all non-plaque microdomains of Tg-m when compared to WT-m.

Next, it was tested whether clasmatodendrosis could also affect the astrocyte clearance of autofluorescent deposits in the *stratum pyramidale* (Figure 5G–J). Three different settings were used to simultaneously acquire autofluorescence and immunostaining signals (Appendix A), and the resulting channels were merged to perform machine learning analysis (MLA) (Figure 5G). As expected, the number of autofluorescent deposits identified by MLA was significantly higher in 6m-Tg and 12m-Tg than in Wt-m (Figure 5H), while no significant difference was found between Wt-m and 2m-Tg-m. However, the rate of association of deposits with APJs was significantly lower in 2m-Tg-m than in WT-m (Figure 5I) and, consequently, the number of non-associated deposits spread in the *stratum pyramidale* was higher (Figure 5J). In the advanced stages of the disease, the number of non-associated deposits progressively increased, being about threefold higher in 12m-Tg-m than in WT-m.

Taken together, these data provided a qualitative and quantitative characterization of the marked clasmatodendrosis occurring in the CA1 hippocampus of Tg-m and how it may significantly hinder astrocyte clearance activity at every stage of the disease. Moreover, it was shown that interactions between APJs and amyloid plaques can also promote intense clasmatodendrosis, concomitant with astrocyte hypertrophy.

### 3.4. Analysis of the In Situ Expression and Localization of GFAP and Cx43 and the Morphometry and Distribution of Cx43 Clusters

In order to assess whether and how the morphological changes described above may be related to alterations in astrocyte connectivity and the integrity of their functional syncytium, confocal analyses were performed on sections immunostained to reveal Cx43 and GFAP. This analysis also involved comparing the expression of Cx43 in APJs of the glia limitans, peri-vascular scar, and peri-plaque scar and observing the interactions between APJs and amyloid deposits within the glial scar. Mixed amyloid/lipofuscin autofluorescence was also obtained by spectral unmixing. Figure 6 shows a quantitative analysis of the mean intensity of Cx43, as well as MLA of Cx43 clusters in glia limitans and perivascular and peri-plaque glial scars. Figure 7 shows representative Z-projections of acquired optical volumes for qualitative assessment of Cx43 localization and APJ interaction with autofluorescent deposits and plaques.

In the *stratum radiatum* of 2m-Tg-m, levels of Cx43 immunofluorescence were lower than in WT-m (Figure 6A and Figure 7A,B), particularly in third and fourth order APJs (Figure 7A insets 1 and 2, 7B insets 3 and 4). In addition, in 6m- and 12m-Tg-m, Cx43 immunofluorescence was lower than in WT-m, but in 12m-Tg-m, the decrease was not significant (Figure 6A). According to previous data [57], astrocytes in 6m- and 12m-Tg-m were characterized by markedly high levels of Cx43 immunofluorescence when contacting amyloid plaques (*stratum radiatum*, Figure 7C panel c1 IN, insets 5 and 6, 7D panel d1 IN, inset 9). Conversely, in non-plaque microdomains of 6m- and 12m-Tg-m, dystrophic APJs showed low levels of Cx43 (Figure 7C panel c2 OUT, insets 7 and 8, 7D panel d2 OUT, inset 10). The above data suggest that in this Tg-m model, there is impairment of the functional syncytium of astrocytes at all stages of the disease in non-plaque microdomains. However, hypertrophic APJs of glial scars associated with blood vessels in 12m-Tgm showed high levels of Cx43 (*stratum radiatum*, Figure 7D panel d2 OUT and inset 11). When compared with normal vessels in WT-m, clusters of Cx43 in these perivascular glial scars of 12m-Tg-m showed greater dispersion around the vessel (Figure 6C, Appendix A) and a different distribution of their volumes, evidenced by a significant reduction in the number of large clusters (>1.5 µm^3^) (Figure 6D) against an equal number of total clusters (Figure 6E). In detail, the large clusters of Cx43 in WT-m were located on the APJs of the glia limitans (Figure 6C first row, Appendix A), possibly involved in their constitutive activities at astrocyte endfeet [58]. Conversely, in perivascular gliosis, numerous small clusters of Cx43 were located on the APJs close to the glia limitans (Figure 6C, second row, Appendix A). Morphometric analyses performed on the APJs surrounding, and not contacting, the amyloid plaques disclosed no significant differences in the localization, total number, and volumetric distribution of Cx43 clusters at these sites, in comparison with the perivascular glial scar (Figure 6C third row, D, E, Appendix A). Moreover, in these areas, the intact APJs displayed a similar net-like arrangement (Figure 7D inset 9a). These data, while supporting the hypothesis of similar roles of peri-plaque and peri-vascular scars, prompted us towards detailed investigations on the interactions between astrocytes and autofluorescent Aβ-deposits within the scar.

Astrocyte meshwork tightly enveloped the amyloid plaques, with numerous thin processes, arranged into dense nets covering the plaque surface (*stratum radiatum*, Figure 7D panel d1 IN, inset 9a, Appendix A, Appendix A) and permeating the neighboring areas (Figure 7D inset 9, Appendix A). Noteworthy, autofluorescence in the hippocampus of 12m-Tg-m mice was predominantly localized in the peri-plaque scars, where numerous conspicuously sized deposits were observed, located radially around the plaques (Figure 7D panel d1 IN, Appendix A). Several merging patterns were observed between the plaques and nearby deposits, the latter being appreciable by their intense autofluorescence. (Figure 7D inner insets in panel d1 IN, Appendix A). All these deposits, both surrounding (Figure 7D inset 9, Appendix A) and fusing with plaques (Figure 7D inner insets in panel d1 IN, Appendix A), adhered tightly to thin APJ meshes, which in turn were connected to the glial envelope of the plaque.

These results indicate that APJs of peri-plaque scar interact with the amyloid plaques surrounding Aβ-peptide deposits and are most likely involved in their fusion.

### 3.5. Analysis of the Expression and Localization of GFAP of Astrocytes and MAP2 of Neurons

In order to ascertain whether all the above-described morphofunctional alterations of the astrocyte meshwork, either reactive or nonreactive, affect the neuronal population, confocal analysis was performed on immunostained sections to reveal GFAP in APJs and MAP2 in dendrites. In addition, nuclei were counterstained with DAPI, and mixed amyloid/lipofuscin autofluorescence was also acquired by Tau Sense photon counting. The interactions of dendrites with amyloid plaques and APJs of the peri-plaque scar were also evaluated. Quantitative MLA-based analyses of the integrated density of MAP2+ fragments around plaques and in equivalent areas within non-plaque microdomains are shown in Figure 6B. Representative z-projections for qualitative assessment of morphology and localization of GFAP+ APJs and MAP2+ dendrites of neurons are shown in Figure 7.

The negligible autofluorescent deposits found in the *stratum pyramidale* of CA1 hippocampus of WT-m were consistently associated with APJs (Figure 8A inset 1a), and their amount and size were lesser than in the equivalent domains of 2m-, 6m- and 12m-Tg-m (Figure 8B inset 2, 8C inset 4, 8D inset 8). Consistent with previous findings [29], numerous GFAP+ fragments were observed in association with the autofluorescent spots in Tg-m (Figure 8B inset 2a, 8C inset 4a, 8D inset 8a). These data confirm the close relationship between overproduction of cytotoxic molecules and clasmatodendrosis in AD. In 6m-Tg-m, amyloid plaques of variable size were detected in large areas of dendrite rarefaction (*stratum radiatum*, Figure 8C panel c1 IN), recalling morphologic patterns of edema. Small, newly-formed plaques were characterized by the occurrence of numerous MAP2+ and GFAP+ fragments on their surfaces (Figure 8C inset 3, Appendix A). Indeed, a significant amount of MAP2+ fragments (Figure 6B, Appendix A) was found in these microdomains of both 6m- and 12m-Tg-m, thus confirming the cytotoxic effects of amyloid plaques. In 12m-Tg-m (*stratum radiatum*, Figure 8D panel d1 IN), the dense meshwork of processes of hypertrophic astrocytes enveloping amyloid plaques was frequently interposed between them and MAP2+ dendrites (Figure 8D insets 5, 6), thus hampering their contacts. The APJs outside of the envelope set up numerous contacts with neighboring dendrites (Figure 8D inset 7).

These data support the hypothesis that peri-plaque glial scarring hinders the interactions between neurons and amyloid plaque.

## 4. Discussion

The data in this study provided consistent support to the working hypothesis, confirming that Aβ deposition may exert early non-reactive effects on astrocytes that, in turn, impair astroglial and microglial clearance activity. The theoretical basis for this hypothesis was provided by previous studies carried out in rat models of CNS aging [10,29].

According to the hypothesis, it was found that astrocytes early interact with Aβ-peptides before the buildup of amyloid plaques and the consequent onset of their reactivity. The data reported here show that this interaction results in clasmatodendrosis and astrocyte meshwork disruption. Such non-reactive modifications are related to an altered Aβ clearance activity of astrocytes as well as microglia, which exhibit a contextual senile-like phenotype and deficits in key migratory processes. These patterns have also been identified in all subsequent stages of the disease, apart from areas occupied by amyloid plaques. In these plaque microdomains, both microglia and astroglia show a typical reactive morphology. Contextually, our data suggest that reactive astrocytes enveloping amyloid plaques within glial scar-like formations are involved in merging processes between Aβ-deposits and, therefore, in plaque build-up.

It is currently accepted that microglial phagocytosis consists of three steps: (i) the “find me”, namely microglial targeting and migrating to proinflammatory molecules; (ii) the “eat me”: phagocytosis of these molecules; (iii) the “digest me”: the endosomal-lysosomal pathway to molecular degradation. Comparison of the expression and cellular localization of two different markers of microglia reactivity (Iba1 and CD68) in transgenic mice allowed differential activation of cellular processes involved in stages (i) and (ii) to be discerned in situ. In 2m-Tg-m, microglia were characterized by increased expression of the phagocytosis marker CD68, compared with WT-m, indicating the onset of reactivity. Numerous factors induce microglial reactivity: neuronal release of chemokines [59,60,61], alarmins [62], ATP [63], and glutamate [64]; metabolic and redox alterations [65,66]; cellular debris and misfolded proteins in the extracellular microenvironment [3]; mechanical alterations of the extracellular matrix ECM [67]. On the other hand, reactive microglia of 2m-Tg-m showed low levels of Iba1 expression and branching, suggesting an impaired “find me” process in phagocytosis. It should be noted that similar features of microglial reactivity were shown in the aged CNS, i.e., inefficient clearance activity [68], despite considerable expression of CD68 [69]. It is worth recalling that low levels of Iba1 and high expression of CD68 have been shown to characterize microglia in age-associated white matter lesions [70]. Moreover, defective branching of microglia and subsequent impairment of their clearance activity is correlated with disassembly of the astrocyte meshwork in the hippocampus of aged rats [10]. The present data indicate that in our model of Aβ deposition, the expression of a senile-like phenotype by microglia, suggestive of altered migration, correlates spatially and temporally with a significant loss of astrocytic meshwork integrity since the early stages of the disease.

Regarding astrocyte functions, it could be expected that in aged rats (22 months), increased deposition of autofluorescent Aβ-peptide on neurons would induce clasmatodendrosis, APJ meshwork disassembly and consequent impairment of astrocytic clearance [29]. However, it was not predictable that similar effects of the Aβ-peptide could be revealed in 2m-Tg-m as well, being relevant to the early increase of Aβ-load in *stratum pyramidale* neurons. In both 6m- and 12m-Tg-m, the presence of amyloid plaques added further complexity, as surrounding astrocytes were characterized by hypertrophic morphology and microglia by elevated expression of Iba1 and CD68. Nonetheless, in non-plaque microdomains, both glial cell populations showed morphological traits and immunohistochemical responses similar to their counterparts in 2m-Tg-m, implying the presence of clasmatodendrosis related impairment of Aβ-clearance also in 6m- and 12m-Tg-m. As a whole, above data suggest that clasmatodendrosis and APJ meshwork disassembly in non-plaque microdomains may favor Aβ-load and, in turn, Aβ-plaque formation at all stages of the disease.

It is known that plaque-microdomains are characterized by specific microenvironmental conditions, which strongly affect the phenotype of microglia [71,72] which releases signaling molecules, e.g., proinflammatory cytokines, and induces astrocyte reactivity [8,73]. This provides a rationale for the known correlation between such induction and the presence of amyloid plaques. Accordingly, the present investigation provides evidence of numerous astrocyte and neuron debris adhering to the plaques, according to a pattern of cell damage and release of proinflammatory molecules and chemoattractants. On the other hand, the present data show that at any stage of the disease, the APJs were usually in contact with amyloid accumulations, ranging from the smaller deposits in 2m-Tg-m to the larger plaques in 12m-Tg-m. Reasonably, these associations between cytotoxic amyloid bodies and related microenvironmental changes can deeply alter astrocyte phenotype and functions in disease progression. This hypothesis is supported by the similarities described here between APJs in peri-plaque and perivascular scars. The latter were frequently observed in the CA1 hippocampus of 12m-Tg-m and are well-known key signatures of the impairment of the blood-brain barrier [74] leading to uncontrolled blood-tissue exchanges and to increased microperfusion dynamics and, in turn, to altered mechanical properties of the tissue. Astrocytes are affected by such environmental changes since they express the Piezo1 mechanosensor, whose involvement in cognitive function was suggested by a recent study in mice [75]. In injured neocortex and spinal cord of rats, atomic force microscopy measurements showed an inverse relation between the stiffness of ECM and astrocyte expression of GFAP, and, even more importantly, tissue consistency at glial scars was softer than in the surrounding area [76]. In this study, we observed morphological patterns of edema with concomitant recovery of branching and GFAP expression by astrocytes in plaque microdomains of 6m- and 12m-Tg-m. Therefore, it can be proposed that local microenvironmental factors, possibly including mechanical stimuli, contribute to promoting the hypertrophic transition of plaque-associated astrocytes, similar to what we observe at the level of the perivascular glial scar. Importantly, peri-plaque and peri-vascular reactive astrocytes shared a further remarkable similarity: both showed high expression of Cx43 assembled in morphometrically comparable clusters, suggesting a similar functional role. It is known that glial scars formed by proliferating astrocytes perform a barrier function in neuropil and blood vessel repair and in regeneration after injury [12]. Indeed, in rat models of ischemic injury, proliferative astrocytes displayed high expression of Cx43, which was suggested to play a role relevant to tissue repair [77]. In this view, present findings support the hypothesis that astrocyte gliosis in plaque microdomains may exert opposite effects. On the one hand, it may result in neuroprotection: peri-plaque APJs meshwork acts as a barrier between Aβ-plaques and dendrites and possibly other tissue components. On the other hand, the recurrent patterns of APJ damage observed at these sites are indicative of the cytotoxic effects that these interactions induce on astrocytes, which may result in the exacerbation of their response, thus eliciting neurodegenerative side effects.

Whether or not amyloid plaques are the main cause of neurodegeneration in AD remains a matter of debate. Nevertheless, clear-cut cytotoxic properties have been attributed to the major components of these plaques, i.e., the amyloid fibrils. Moreover, these accumulations of harmful Aβ-peptides and cell debris are known to evoke major inflammatory responses from both astrocytes and microglia, being therefore expected to play a main role in the development of AD. The present data help to elucidate this issue, introducing a possible pivotal role of astrocytes in human AD. Indeed, in all stages of the disease, distinct interactions were found between APJs and autofluorescent deposits throughout the CA1 hippocampus. These deposits were obvious in the *stratum pyramidale* from the early stages of the disease, and in the late stages, they were also clustered within the peri-plaque glial scars, closely related to the APJ meshwork. Such autofluorescent dense deposits radially surrounded the plaques or merged with them. The involvement of APJs in merging processes between deposits and plaques has also been observed. In consideration of these findings, we propose a novel aspect of astrocyte reactivity to the overproduction of Aβ-peptides in AD: reactive astrocytes in the peri-plaque scar are involved in the clusterization of Aβ-deposits spread in the nervous tissue, thus contributing to amyloid plaque build-up. Since a direct correlation links the surface area and the rate of exchange processes between cells and environmental components, it can be assumed that the cytotoxic interactions between Aβ peptides and CNS cells, including astrocytes, are lesser when peptides are clustered in amyloid plaques rather than dispersed throughout the nervous tissue. This hypothesis sheds light on relevant processes underlying amyloid plaque buildup and proposes for these structures in AD a role of repositories for cytotoxic molecules isolated from nerve tissue by glial cells. In the interaction framework, astrocytes may partially compensate for the inefficiency of microglia to clear Aβ-peptides. Indeed, the accumulation of possible targets in microenvironmental domains characterized by high concentrations of chemotactic molecules may favor microglia to complete the “find me” step of phagocytosis.

## 5. Conclusions

Overall, this study indicates that in the Tg-m model of AD, different environmental challenges occurring in distinct microenvironmental domains and/or stages of the disease can alter the phenotype of glial cells, either by inducing non-reactive effects or by eliciting and modulating reactive responses. In early, intermediate, and late stages of AD progression, non-reactive effects of amyloid fibril overproduction on astrocytes consisted of clasmatodendrosis and severe disruption of APJ meshwork. In turn, disruption of this meshwork was accompanied by decreased interactions of APJs with cytotoxic molecules in the *stratum pyramidale* and shifting of microglia reactivity towards a senile-like phenotype, thus hindering Aβ-peptide clearance. Impaired clearance of Aβ peptides may be counterbalanced by novel functions of glial cells: amyloid plaques may be considered as repositories of cytotoxic Aβ, gathered and segregated by hypertrophic, reactive astrocytes within glial scars. In this view, the wide range of phenotypes shown by astrocytes and microglia in AD models may result from continuous adaptations to dynamically changing microenvironmental conditions. These adaptations may have defective aspects, prevalently referred to as the “degenerative” phenotypes of reactive astrocytes and microglia. Indeed, unfitting mechanisms of glial cell reactivity could not be filtered out by natural selection since neurodegenerative diseases such as AD usually involve late, generally non-reproductive phases of the human lifespan. However, our data, as well as recent literature, point to a different perception of the intrinsic complexity underlying the phenotypic alterations of glial cells, whether reactive or non-reactive, in the progression of AD, thus tempering the concept of a glial cell shift from a “neuroprotective” to a “neurodegenerative” phenotype. In this perspective, present data emphasize the role played by microenvironmental alterations in glial cell phenotypic changes and provide an adequate conceptual frame to understand the consequent perturbations in functions and reciprocal inductions of microglia and astroglia in AD.

## Figures and Tables

**Figure 1 cells-12-02258-f001:**
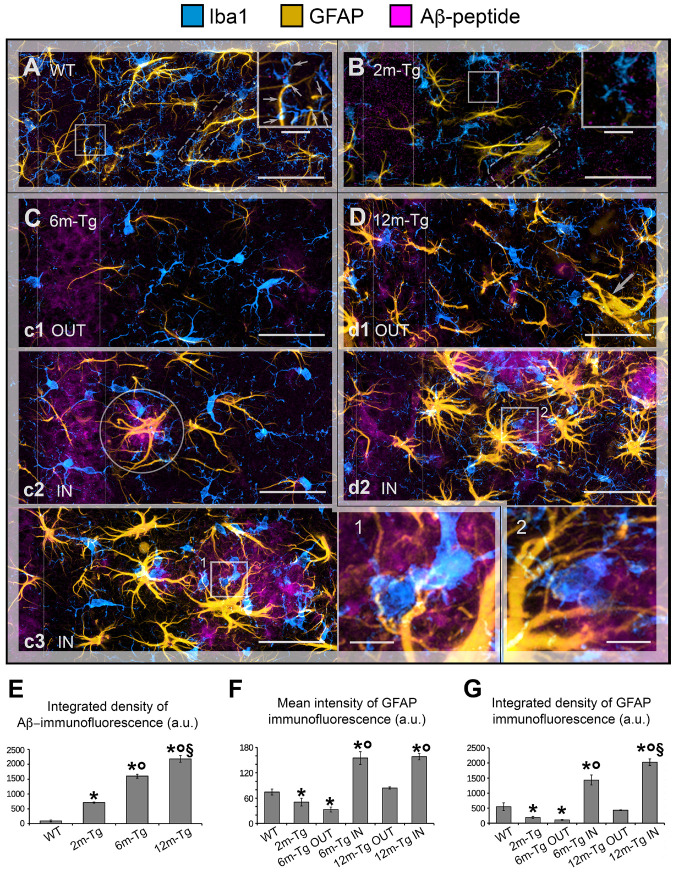
(**A**–**D**) Comparison between the features of microglial reactivity and astrocyte modifications, both reactive and non-reactive, in the transgenic mouse hippocampus. Hippocampal sections of in wild type mice (WT-m) (**A**), and 2 months (2m; **B**), 6 months (6m; **C**) and 12 months (12m; (**D**) old TgCRND8 mice (Tg-m) were stained to reveal microglia Iba1 (cyan), astrocyte glial fibrillary acidic protein (GFAP; yellow), and amyloid β (Aβ) peptide; magenta). (**A**) Low magnification z-projections of the CA1 in WT-m; inner inset in A shows magnified details of contacts (thin arrows) between microglia branches and APJs. (**B**) Low magnification z-projection of the CA1 in 2m-Tg-m; inner inset in B shows magnified details of contacts (thin arrows) between microglia branches and astrocyte processes APJs. APJs involved in the formation of glia limitans perivascularis are shown in (**A**,**B**) (dashed box). (**C**) Low magnification z-projection of the CA1 in 6m-Tg-m. Panels (**c1**–**c3**) are representative images of astrocytes and microglia in non-plaque (**c1**) and in plaque- (**c2**,**c3**) microdomains. Small-sized plaque is schemed in (**c2**). Inset 1 shows magnified details of patterns of microglial phagocytosis on plaques. (**D**) Low magnification z-projection of the CA1 in 12m-Tg-m. Panels (**d1**,**d2**) are representative images of astrocytes and microglia in non-plaque (**d1**) and plaque- (**d2**) microdomains; the arrow in panel d1 indicates hypertrophic APJs involved in the formation of a perivascular glial scar. Inset 2 shows details of patterns of microglial phagocytosis of plaques. Vertical dotted lines delimit the *stratum pyramidale*. Bars: (**A**,**B**,**c1**–**c3**,**d1**,**d2**) = 35 µm; 1,2 = 5 µm; inner insets = 11 µm. (**E**–**G**) Immunofluorescence intensity of Aβ-peptide in the *stratum pyramidale* (**E**) and GFAP in the CA region (**F**,**G**) of WT-m, and 2m-, 6m- and 12m-Tg-m. All data were obtained from 5 mice per experimental group and are expressed as mean ± SE intensity of immunofluorescence. (**E**) Integrated density of Aβ-immunofluorescence (F(3,16) = 1683, * *p* < 0.05 vs. WT, ° *p* < 0.01 vs. 2m-Tg, ^§^ *p* < 0.05 vs. 6m-Tg). ((**F**,**G**) Mean Intensity (**F**) and integrated density (**G**) of GFAP immunofluorescence: (**E**)) F(5,19) = 43, * *p* < 0.05 vs. WT, ° *p* < 0.001 vs. 2m-Tg; (**G**) F(5,19) = 63.2, * *p* < 0.05 vs. WT, ° *p* < 0.01 vs. 2m-Tg, ^§^ *p* < 0.001 vs. 6m-Tg IN. All statistical analyses were performed by one-way ANOVA followed by Tukey *post hoc* tests.

**Figure 2 cells-12-02258-f002:**
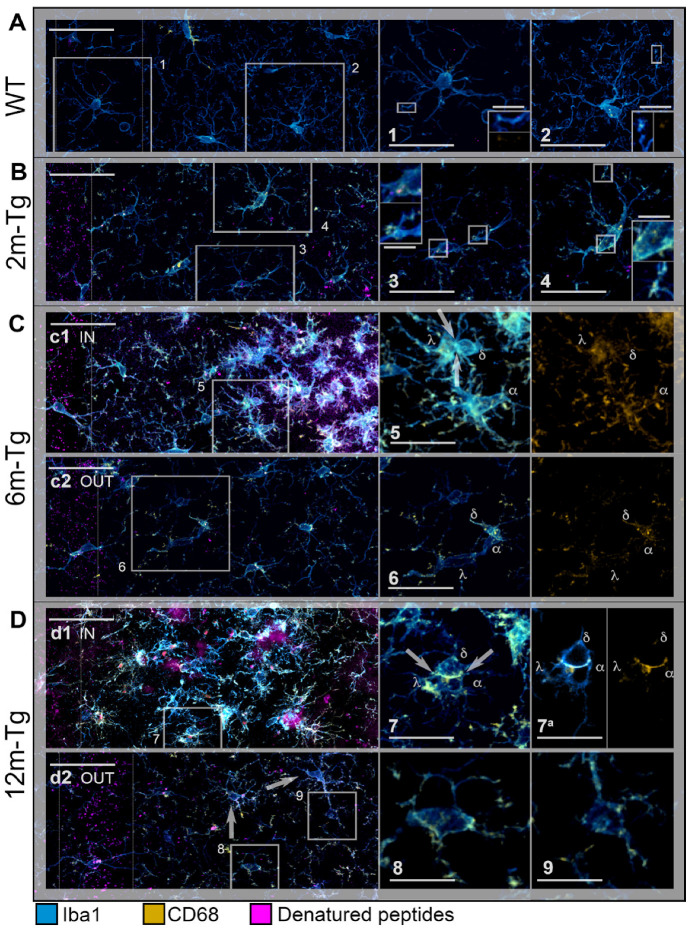
Reactivity of microglia in the hippocampus of transgenic mice (TgCRND8). Microglia in CA1 hippocampus of WT-m (**A**), 2m- (**B**), 6m- (**C**), and 12m- (**D**) Tg-m were immunostained to reveal Iba1 (cyan) and CD68 (yellow); autofluorescence from mixed lipofuscin/Aβ-peptides is also shown (denatured peptides, magenta). (**A**) Low magnification z-projections of the CA1 in WT-m; insets 1 and 2 show details of the corresponding areas in A, inner insets in 1 and 2 show magnified details of expression of Iba1 (upper inset in 1, left inset in 2) and CD68 (lower inset in 1, right inset in 2) on microglial branches. (**B**) Low magnification z-projections of the CA1 in 2m-Tg-m; insets 3 and 4 show details of the corresponding areas in B; inner insets 3 and 4 show magnified details of expression of Iba1 and CD68 on the cell body (upper insets) and branches (lower insets) of microglia. (**C**) Low magnification Z-projection of the CA1 in 6m-Tg-m. Panel (**c1**) shows microglia crowding around large deposits of autofluorescence (IN); panel (**c2**) shows microglia far from the autofluorescent deposit (OUT). Insets 5 and 6 show magnified details of the corresponding areas in (**c1**) (5) and (**c2**) (6): merged Iba1 and CD68 immunofluorescence (left panel) and CD68 immunofluorescence alone (right panel) are shown; λ, α, δ indicate three distinct microglial cells; arrows in inset 5 indicate soma-soma adhesion between microglia. (**D**) Low magnification z-projection of the CA1 in 12m-Tg-m. Panel (**d1**) shows microglia surrounding large deposits of autofluorescence (IN); arrows in inset 7 indicate soma-soma adhesion between microglia. Panel (**d2**) shows microglia away from the autofluorescent deposits (OUT); arrows indicate branched microglia. Insets 7 and 9 show magnified details of the corresponding areas in (**d1**) (7) and (**d2**) (8,9); λ, α, δ indicate three distinct microglial cells. Inset 7a shows merged Iba1 and CD68 immunofluorescence (left panel) and CD68 immunofluorescence alone (right panel) in a single optical section of these cells; nuclei correspond to the inner circular unstained areas. Autofluorescence was omitted in all insets. Vertical dotted lines delimit the *stratum pyramidale*. Bars: (**A**,**B**,**c1**,**c2**,**d1**,**d2**) = 35 µm; 1–9 = 10 µm; inner insets = 4 µm.

**Figure 3 cells-12-02258-f003:**
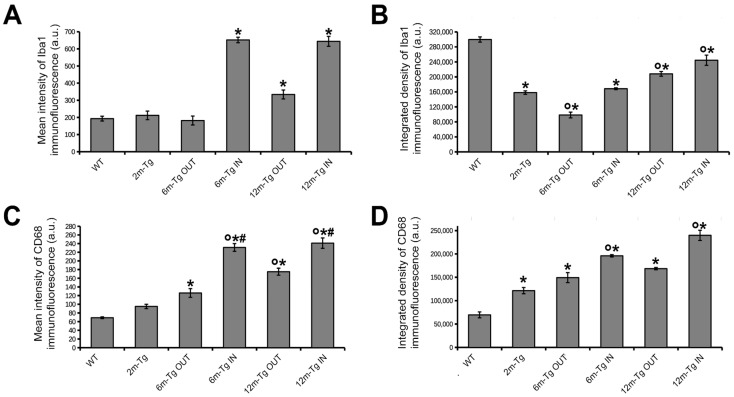
Expression of Iba1 and CD68 in the hippocampus of transgenic mice. Mean intensity (**A**,**C**), and integrated density (**B**,**D**) of Iba1 and CD68 immunofluorescence were measured in the hippocampus of WT-m, and 2m-, 6m-, and 12m-Tg-m. Values are expressed as mean ± SE. All statistical analyses were performed by One Way ANOVA followed by Tukey post hoc tests (*n* = 3): (**A**) F(5,12) = 269, * *p* < 0.001 vs. WT; (**B**) F(5,12) = 69.7, * *p* < 0.05 vs. WT, ° *p* < 0.05 vs. 2m-Tg; (**C**) F(5,12) = 88.4, * *p* < 0.01 vs. WT, ° *p* < 0.001 vs. 2m-Tg, # *p* < 0.001 vs. 6m-Tg OUT; (**D**) F(5,12) = 743, * *p* < 0.001 vs. WT, ° *p* < 0.001 vs. 2m-Tg.

**Figure 4 cells-12-02258-f004:**
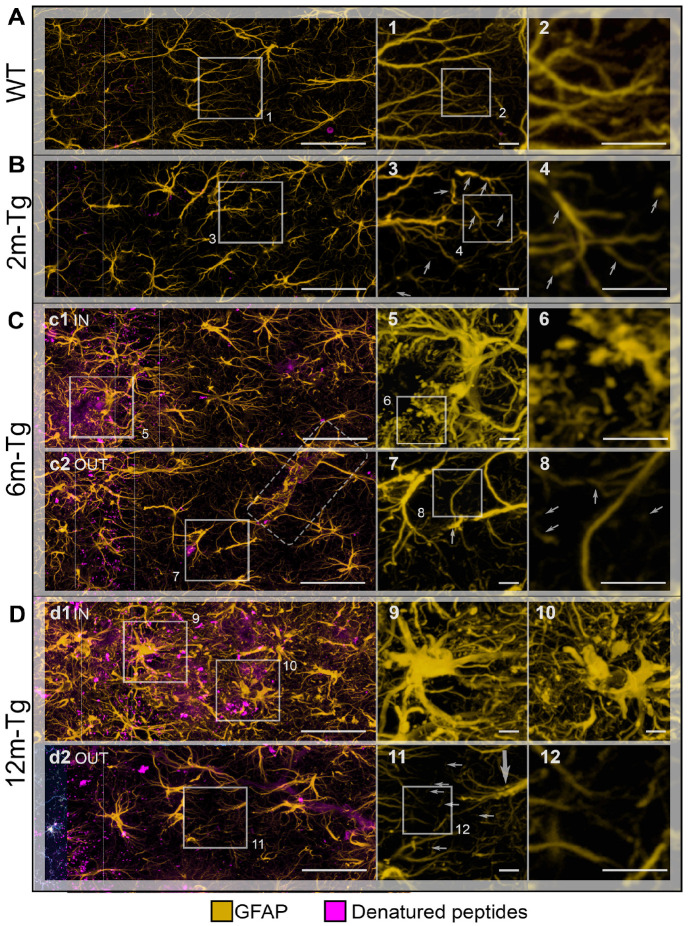
Functional integrity of astrocyte meshwork in transgenic mouse hippocampus: morphological features. Astrocytes in the CA1 hippocampus of WT-m (**A**), and 2m- (**B**), 6m- (**C**), and 12m- (**D**) Tg-m were stained to reveal cytoskeletal GFAP (yellow); autofluorescence of Aβ peptides is also shown (denatured peptides, magenta). (**A**) Low magnification z-projections of the CA1 in WT-m; insets 1 and 2 show magnified details of the corresponding areas in A (1) and 1 (2). (**B**) Low magnification z-projection of the CA1 in 2m-Tg-m; insets 3, 4 show magnified details of the corresponding areas in B (3) and 3 (4). (**C**) Low magnification z-projection of the CA1 in 6m-Tg-m. Panel (**c1**) shows astrocytes in a plaque-microdomain; panel (**c2**) shows astrocytes in a non-plaque microdomain; APJs involved in the formation of blood-brain barrier are framed in c2. Insets 5–8 show magnified details of the corresponding areas in c1 (5), 5 (6), c2 (7) and 7 (8). (**D**) Low magnification z-projection of the CA in 12m-Tg-m. Panel (**d1**) (IN) shows astrocytes in a plaque microdomain; panel (**d2**) shows astrocytes in a non-plaque microdomain. Thin arrows indicate GFAP+ fragments; the thick arrow points to a spiraliform primary process. The vertical dotted lines delimit the *stratum pyramidale*. APJs involved in the formation of the blood-brain barrier are framed in (**C**). Insets 9–12 show magnified details of the corresponding areas in (**d1**) (9,10), (**d2**) (11), and 11 (12). Bars: (**A**,**B**,**c1**,**c2**,**d1**,**d2**) = 33 µm; 1–12 = 4 µm.

**Figure 5 cells-12-02258-f005:**
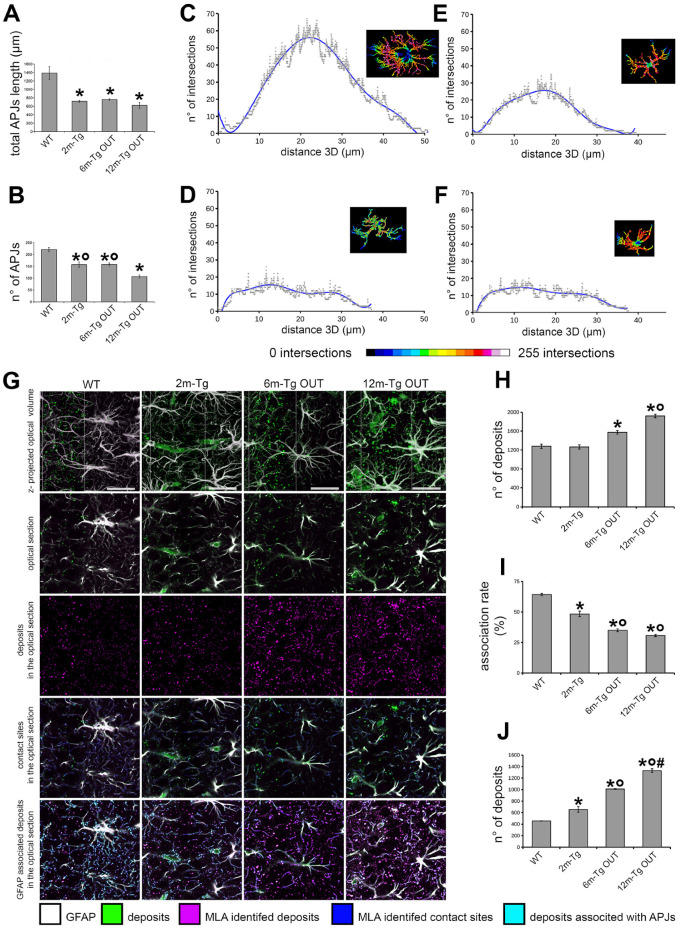
Quantitative analysis of the functional integrity of the astrocyte meshwork in the hippocampus of transgenic mice. Sections of the hippocampus from WT-m, and 2m-, 6m- and 12m-Tg-m were stained for GFAP in astrocytes. APJs were traced by the ImageJ plugin “Simple Neurite Tracer” on 3D optical volumes (about 36,800 µm^3^). (**A**) Total length of APJs (F(2,6) = 16.5, * *p* < 0.01 vs. WT). (**B**) Number of APJs that were neither engaged in connections with amyloid plaques nor in the blood-brain barrier framework (F(2,6) = 31.97, * *p* < 0.01 vs. WT, ° *p* < 0.05 vs. 12m-Tg. All statistical analyses were performed by One-way ANOVA followed by Tukey *post hoc* tests. (**C**–**F**) The Sholl analyses were performed on representative astrocytes neither involved in amyloid plaques nor in the blood-brain barrier framework (Appendix A), from WT-m, and 2m- (**D**), 6m- (**E**) and 12m- (**F**) Tg-m. Linear plots show the number of intersections between APJs and Sholl’s spheres at increasing distances from the center of the cell. Inner insets show volume renderings of the analyzed APJs, and the color scale provides topographic indications of the number of intersections at different distances from the center of the cell. (**G**–**J**) MLA of the association of fluorescent deposits with APJs in the *stratum pyramidale* of transgenic mice. (**G**) The 495–525_l458_ (magenta color scale) and 560–590_l458_ (green color scale) channels were merged to allow discrimination of APJs (white) and autofluorescent deposits (green) in the z-projections of the optical volumes (1st row) and individual optical sections (2nd row). The channel of autofluorescent deposits (magenta) obtained by MLA is shown in the 3rd row. Contacts (blue) between autofluorescent deposits and APJs (Gray) are shown in 4th row; autofluorescent deposits associated with APJs (cyan) are shown in 5th row. (**H**–**J**) Quantitative evaluation of the total number of autofluorescent deposits (**H**), their association rate with APJs (**I**), and the number of non-associated deposits (J) in the CA1 *stratum pyramidale* of WT-m, and 2m-, 6m- and 12m-Tg-m. (**H**) F(3,8) = 47.1, * *p* < 0.01 vs. WT, ° *p* < 0.01 vs. 6m-Tg OUT. (**I**) F(3,8) = 155, * *p* < 0.001 vs. WT, ° *p* < 0.01 vs. 3m-Tg. (**J**) F(3,8) = 2032, * *p* < 0.01 vs. WT, ° *p* < 0.001 vs. 3m-Tg, # *p* < 0.001 vs. 6m-Tg OUT. Bars, G = 25 µm.

**Figure 6 cells-12-02258-f006:**
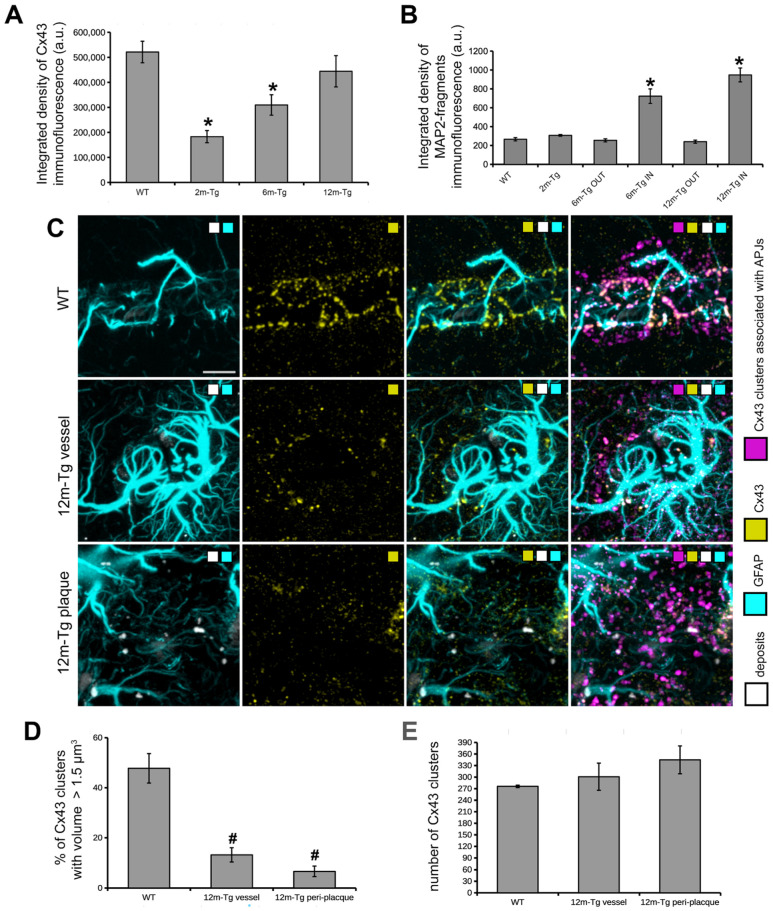
Functional integrity of the astrocyte meshwork and dendrite fragmentation in transgenic mouse hippocampus. (**A**) Quantitative estimation of integrated density of Cx43 immunofluorescence; F(3,8) = 11.8, * *p* < 0.01 vs. WT, *n* = 3. (**B**) Quantitative estimation of integrated density MAP2+ fragments; F(5,18) = 30.9, * *p* < 0.001 vs. WT, *n* = 4. Values are expressed as mean ± SE. (**C**) Astrocyte GFAP (cyan color scale) and Cx43 (yellow color scale) immunostaining and autofluorescent deposits (gray color scale) are shown in blood vessels of WT-m (1st row), and perivascular (2nd row) and peri-plaque (3rd row) glial scars. Clusters Cx43 on APJs were identified by MLA (magenta color scale). (**D**,**E**) Quantitative evaluation of the percentage of clusters showing a volume size > 1.5 µm^3^ (**D**) and the total number of Cx43 clusters (**E**). (**D**) F(2,6) = 18.5, # *p* < 0.01 vs. WT. All statistical analyses were performed by One-way ANOVA followed by Tukey post hoc tests. Bars: C = 10 µm.

**Figure 7 cells-12-02258-f007:**
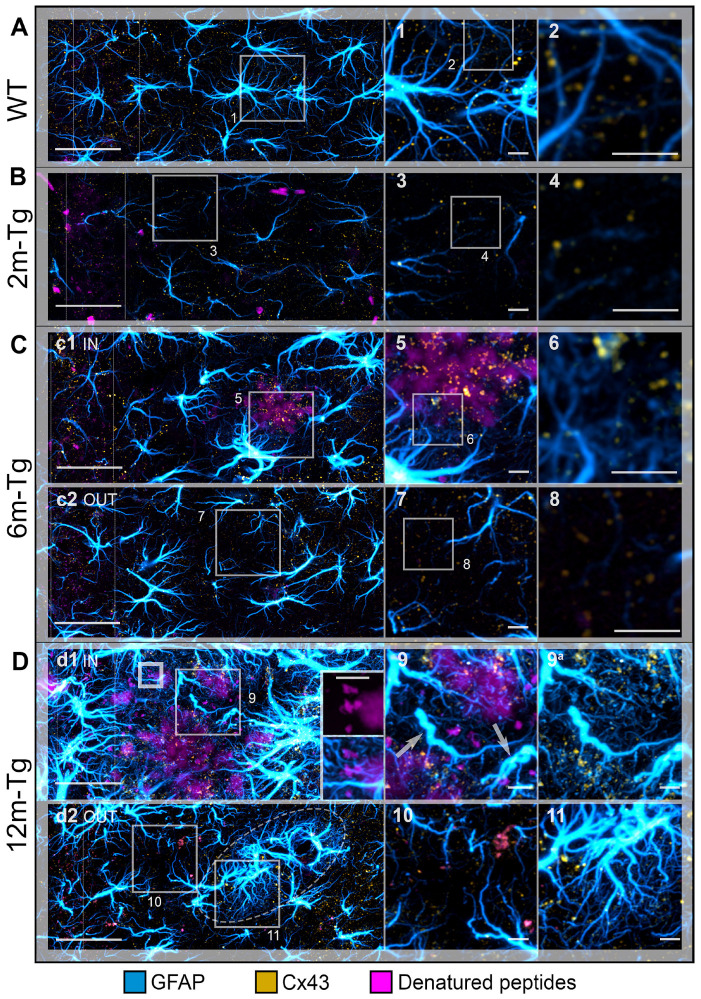
Functional integrity of astrocyte meshwork in the hippocampus of transgenic mice: Cx43 localization. Astrocytes in the CA1 hippocampus of WT-m (**A**), and 2m- (**B**), 6m- (**C**), and 12m- (**D**) Tg-m were immunostained to reveal GFAP (cyan); autofluorescence of Aβ peptides (magenta). (**A**) Low magnification z-projections of the CA1 in WT-m; insets 1 and 2 show details of the corresponding areas in A (1) and 1 (2). (**B**) Low magnification z-projections of the CA1 in 2m-Tg; insets 3 and 4 show details of the corresponding areas in B (3) and 3 (4). (**C**) Low magnification z-projections of the CA1 in 6m-Tg-m. Panels c1 and c2 show astrocytes in peri-plaque (**c1**) and non-plaque (**c2**) microdomains. Insets 5–8 show details of the corresponding areas in c1 (5), 5 (6), c2 (7), and 7 (8); in inset 6, only Cx43 and GFAP immunofluorescence are shown. (**D**) Low magnification z-projections of the CA1 in 12m-Tg-m. Panels (**d1**,**d2**) show astrocytes in peri-plaque (**d1**) and non-plaque (**d2**) microdomains. The inner inset in (**d1**) is a magnified detail of the small-framed area showing merging of small deposits and amyloid plaques (magenta channel, upper) and their interactions with APJs (magenta and cyan channels, lower). Elliptical frame in (**d1**) delimits a perivascular glial scar in both tangential view and transversal section. Insets 9–11 show details of the corresponding areas in (**d1**) (9) and (**d2**) (10, 11); inset 9 shows immunofluorescence of GFAP and Cx43 merged with (9) or without (9a) the autofluorescent signal; arrows point at spiraliform APJs. Vertical dotted lines delimit the *stratum pyramidale*. Bars: (**A**,**B**,**c1**,**c2**,**d1**,**d2**) = 33 µm; 1–11 = 4 µm; inner insets = 5 µm.

**Figure 8 cells-12-02258-f008:**
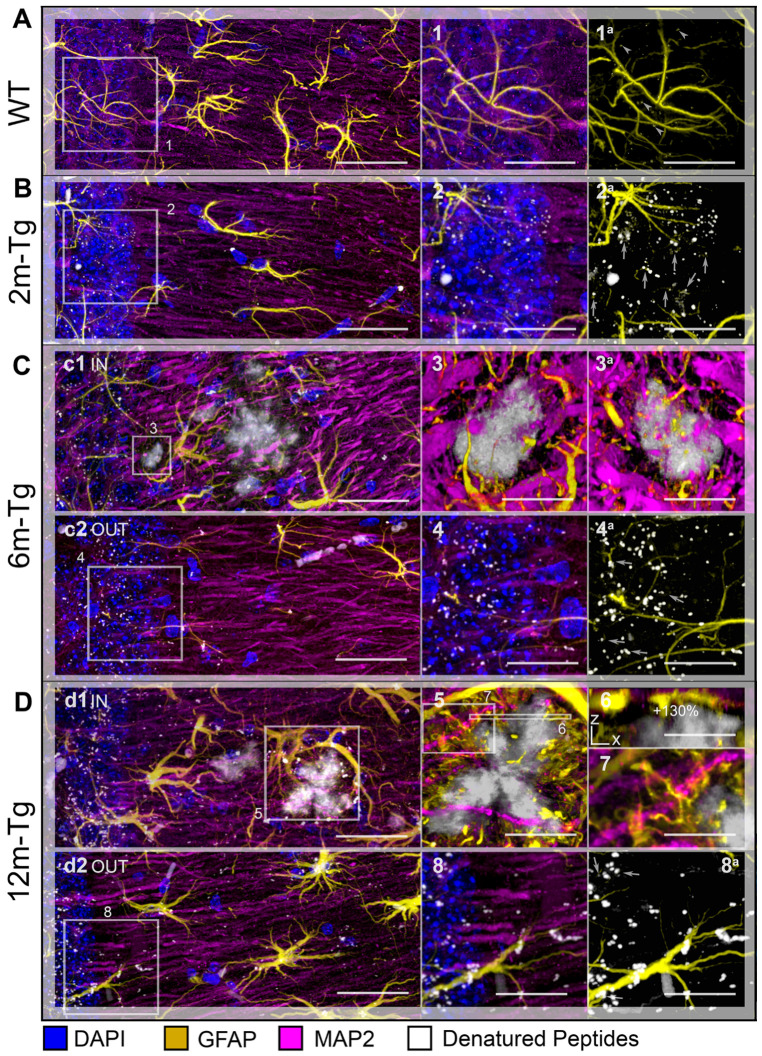
Interactions of the APJ meshwork with denatured peptides in the hippocampus of transgenic mice: effects on hippocampal neurons. Hippocampal sections of WT-m (**A**), and 2m- (**B**), 6m- (**C**), and 12m- (**D**) Tg-m were immunostained to reveal astrocyte GFAP (yellow), neuron MAP2 (magenta); the nuclei were counterstained with DAPI (blue); autofluorescence from mixed lipofuscin/Aβ-peptides is also shown (gray). (**A**) Low magnification z-projections of the CA1 in WT-m. Insets 1 and 1a show details of the corresponding areas in A, with (1) and without (1a) MAP2 and DAPI; arrowheads indicate autofluorescent deposits. (**B**) Low magnification z-projections of the CA1 in 2m-Tg-m. Insets 2 and 2a show details of the corresponding areas in B, with (2) and without (2a) MAP2 and DAPI. (**C**,**D**) Low magnification z-projections of the CA1 in 6m-Tg-m (**C**) and 12m-Tg-m (**D**); APJs and dendrites are shown in microdomains near (panels (**c1**,**d1**)), and far from the plaques (panels (**c2**,**d2**)). Inset 3/3a is a 3D volume rendering of a small plaque highlighted in panel (**c1**): upper (3) and lower (3a) surfaces of the plaque are shown; insets 4–8 show details of the corresponding areas in (**c2**) (4, 4a), (**d1**) (5), 5 (6, 7) and (**d2**) (8, 8a); inset 6 is a Y-projection of the XZ planes obtained by a reslice of the original optical volume. Small arrows indicate patterns of APJ fragmentation. Bars: (**A**,**B**,**c1**,**c2**,**d1**,**d2**) = 25 µm; 1–8 = 6 µm.

**Table 1 cells-12-02258-t001:** List of the primary antibodies used in this study.

Antibody	Specificity	Supplier, Source, Dilution
Iba1 Ionized calcium-binding adaptor molecule 1	A calcium-binding protein with actin-bundling activity that participates in branching, membrane ruffling, and phagocytosis in reactive microglia	WAKO (Osaka, Japan) Code #016-20001 rabbit, 1:300
CD68 Cluster of Differentiation 68	This marker plays a role in phagocytic activity of tissue macrophages, both in intracellular lysosomal pathways and in extracellular cell–cell and cell-pathogen interactions. It mediates the homing of macrophage subsets to particular tissue sites	AbCam (Cambridge, UK) Code #Ab955 rabbit, 1:100
Cx43 Connexin 43	Structural protein of gap junctional connexons allowing ion and small molecule transit between neighboring cells. It is involved in the onset and maintenance of astroglial functional syncytium	Cell Signaling (Danvers, MA, USA) Code #3512 rabbit, 1:50
GFAP Glial Fibrillary Acidic Protein	A class-III intermediate filament and astrocyte-specific marker expressed mainly in APJs	DakoCytomation (Glostrup, Denmark) Code #Z0334 mouse, 1:500
Aβ Beta-amyloid	A 36–43 amino acid peptide, the main component of amyloid plaques found in the CNS of AD people and related animal models	Covance (Emeryville, CA, USA) Code #SIG-39320, mouse, 1:400
MAP2 Microtubule-associated protein 2	Neuron-specific proteins of the neuro-tubular cytoskeleton widely used to identify neuronal cells and trace their processes	Cell Signaling (Danvers, MA, USA) Code #4542 rabbit, 1:100

## Data Availability

The data presented in this study are available on request from the corresponding author.

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
