# Peer review of "Morphofunctional Investigation in a Transgenic Mouse Model of Alzheimer’s Disease: Non-Reactive Astrocytes Are Involved in Aβ Load and Reactive Astrocytes in Plaque Build-Up"

_cells, 2023, doi:10.3390/cells12182258_

Round 1

Reviewer 1 Report (Previous Reviewer 2)

The authors have given a satisfactory explanation of the questions raised. The over all scope and aim have not improved significantly, yet the presented data and interpretation seems satisfactory.

Reviewer 2 Report (New Reviewer)

There is nothing new here, by the paper is solid and adds to our knowledge regarding the role of astrocytes in Alzheimer’s pathology. Thus, I would suggest acceptance.

none

This manuscript is a resubmission of an earlier submission. The following is a list of the peer review reports and author responses from that submission.

Round 1

Reviewer 1 Report

The authors use confocal microscopy to  investigate the morphological changes in neurons, microglia, and astrocytes in a disease mouse model of Alzheimer’s. The authors compare three ages 2, 6, and 12 months. The study of cross talk between all three cell types is an important area of research. The use of microscopy to show activation of microglia and astrocytes is well-established. And the cross talk between astrocytes, microglia, neurons are well known. See Matejuk et. al. Crosstalk Between Astrocytes and Microglia: An Overview 2020 [1]. Enthusiasm was decreased by a lack of justification beyond what is already known. The paper would benefit from editing from a native English speaker.

Point #1. Before each set of experiments, the authors need to say what is not known and how confocal of neurons, astroglia, and microglia together, reveal something that isn’t already known.

Point #2 Its unclear how confocal alone can show astrocytes promote Ab load and plaque buildup through a reactive modification. The authors observe morphological changes, but correlation isn’t causation.  The whole paper needs to be better structured around 1) what can confocal tell us that isn’t already known about these AD mouse brains, and 2) What are the original findings and why are they significant.  

Point #3. There is substantial research showing that lipids are involved in the communication between astrocytes and microglia (Wang et al Neuron 2021) [2] and astrocytes and neurons (Wang et al PNAS 2021)[3]. And astrocytes release toxic lipids (Guttenplan et. al. Nature 2021) [4]. Given that apoE is the most common genetic marker for sporadic AD, this mechanism warrants discussion.

1.        Matejuk, A.; Ransohoff, R.M. Crosstalk Between Astrocytes and Microglia: An Overview. Front. Immunol. 2020, 11, 1–11, doi:10.3389/fimmu.2020.01416.

2.        Wang, C.; Xiong, M.; Gratuze, M.; Bao, X.; Shi, Y.; Andhey, P.S.; Manis, M.; Schroeder, C.; Yin, Z.; Madore, C.; et al. Selective Removal of Astrocytic APOE4 Strongly Protects against Tau-Mediated Neurodegeneration and Decreases Synaptic Phagocytosis by Microglia. Neuron 2021, 1–18, doi:10.1016/j.neuron.2021.03.024.

3.        Wang, H.; Kulas, J.A.; Wang, C.; Holtzman, D.M.; Ferris, H.A.; Hansen, S.B. Regulation of Beta-Amyloid Production in Neurons by Astrocyte-Derived Cholesterol. Proc. Natl. Acad. Sci. 2021, 118, e2102191118, doi:10.1073/pnas.2102191118.

4.        Guttenplan, K.A.; Weigel, M.K.; Prakash, P.; Wijewardhane, P.R.; Hasel, P.; Rufen-Blanchette, U.; Münch, A.E.; Blum, J.A.; Fine, J.; Neal, M.C.; et al. Neurotoxic Reactive Astrocytes Induce Cell Death via Saturated Lipids. Nature 2021, doi:10.1038/s41586-021-03960-y.

Reviewer 2 Report

1.       The study will benefit if the authors include a summary figure depicting changes in microglia, astrocyte interaction at different ages in AD mouse models.

2.       Authors discuss that mechanical properties of ECM could be detected by astrocytes to form APJs. The authors should stain for mechanosensors in astrocytes for example- Piezo1.

3.       The authors should explain the significance of using CX43 in further details.

4.        The authors should address functional relevance of clasmatodendrosis observed w.r.t BBB breakdown, astrocyte endfeet or interaction with endothelial cells. CO-staining the sections with BBB markers, AQP4 etc. would be useful in addition to morphological features of astrcoytes.